# Nanosecond pulsed electric signals can affect electrostatic environment of proteins below the threshold of conformational effects: The case study of SOD1 with a molecular simulation study

**Elena della Valle[1], Paolo Marracino[2], Olga Pakhomova[3], Micaela Liberti[4], Francesca Apollonio[4]***

**1** BioElectronic Vision Lab, University of Michigan, Ann Arbor, Michigan, United States of America, **2** Rise Technology S.r.l., Rome, Italy, **3** Frank Reidy Research Center for Bioelectrics, Old Dominion University, Norfolk, Virginia, United States of America, **4** Department of Information Engineering, Electronics and Telecommunications, Sapienza University of Rome, Rome, Italy

☯ These authors contributed equally to this work.
* francesca.apollonio@uniroma1.it

**Data Availability Statement:** All necessary files needed to produce MD trajectories and perform post-elaborations are available through figshare

## Abstract

Electric fields can be a powerful tool to interact with enzymes or proteins, with an intriguing perspective to allow protein manipulation. Recently, researchers have focused the interest on intracellular enzyme modifications triggered by the application of nanosecond pulsed electric fields. These findings were also supported by theoretical predictions from molecular dynamics simulations focussing on significant variations in protein secondary structures. In this work, a theoretical study utilizing molecular dynamics simulations is proposed to explore effects of electric fields of high intensity and very short nanosecond duration applied to the superoxide dismutase (Cu/Zn-SOD or SOD-1), an important enzyme involved in the cellular antioxidant defence mechanism. The effects of 100-nanosecond pulsed electric fields, with intensities ranging from $10^8$ to $7 \times 10^8$ V/m, on a single SOD1 enzyme are presented. We demonstrated that the lowest intensity of $10^8$ V/m, although not inducing structural changes, can produce electrostatic modifications on the reaction centre of the enzyme, as apparent from the dipolar response and the electric field distribution of the protein active site. Electric pulses above $5 \times 10^8$ V/m produced a fast transition between the folded and a partially dena-tured state, as inferred by the secondary structures analysis. Finally, for the highest field intensity used ($7 \times 10^8$ V/m), a not reversible transition toward an unfolded state was observed.

## Introduction

The Superoxide is generated by many life processes, which include aerobic metabolism, oxida-tive phosphorylation, and photosynthesis, respiratory burst in the immune response of

with the following DOI: (https://doi.org/10.6084/m9.figshare.9642638.v1). Also the modified versions of sim_util.c (for both the MP and BP signals) library have been included. It is necessary to recompile the gromacs package once the original simu_util.c file is overwritten with the ones provided.

**Funding:** Rise Technology srl provided support in the form of salaries for the author [P. Marracino], but did not have any additional role in the study design, data collection and analysis, decision to publish, or preparation of the manuscript. The specific roles of this author are articulated in the 'author contributions' section.

**Competing interests:** Rise Technology srl affiliation does not alter our adherence to all PLOS ONE policies on sharing data and materials.

stimulated macrophages and neutrophils [1]. Superoxide-dependent formation of hydroxyl radicals (including the superoxide ones) is essential in oxygen toxicity [2, 3]. The reactive oxygen species (ROS) can cause an inflammation and cell and tissue injuries accompanied by DNA damages [4, 5]. ROSs lay background for the arising of human pathologies, including ischemic reperfusion injury, cardiovascular disease, cancer, aging and neurodegenerative diseases [6, 7].

The enzyme superoxide dismutase (SOD) takes part to the cellular antioxidant defence mechanism, inactivating the superoxide radical $O_2^-$ [8, 9] at some of the fastest enzyme rates known. It essentially acts as a master key, controlling cellular ROS levels, with potential use as therapeutic agents in oxidative stress-related diseases [10–12].

Specifically, the Cu/Zn-SOD (or SOD1) [13] is a homodimeric metalloenzyme containing in each subunit a catalytic copper (Cu) and zinc (Zn) metal ions that are essential for dismutation reaction of the superoxide radical $O_2^-$. To prevent the accumulation of the $O_2^-$ radical, in the first step of the dismutation reaction, the $O_2^-$ is oxidized by $Cu^{2+}$ to molecular oxygen ($O_2$) and subsequently a second superoxide anion is reduced by $Cu^+$ to produce hydrogen peroxide ($H_2O_2$).

Recent studies experimentally investigated the role of the SOD1 in ALS (Amyotrophic lateral sclerosis), finding an alteration of the glutamate release in the mice spinal cord [14], while other ones found abnormal expression of the SOD1 in patients affected by neuronal disorders [15, 16]. Moreover, it has been found [17, 18] that the accumulation of the $O_2^-$ radical, causing oxidative stress, is implicated in neurodegenerative diseases as is the ALS [19] or in the Parkinson disorder [20].

All these findings evidence an important role of the SOD1 dismutation reaction as well as of the diffusion rate of $O_2^-$ nearby the SOD1 reaction centre. The structure of the SOD1 active site reveals important features as hydrogen-bonding and metal-binding motifs making possible a mechanism known as 'electrostatic guidance' [21, 22], that promotes dismutation reaction with time scales faster than $O_2^-$ diffusion rate. Such a mechanism, firstly proposed by Getzoff and colleagues in 1983 [13], hypothesizes that the arrangement of electrostatic charges around the active site of SOD1 promotes productive enzyme-substrate interaction through substrate guidance and charge complementarity.

Electrostatic guidance is a highly susceptible mechanism since the arrangement of charges at the active site is dependent on the morphology and dynamics of the overall enzyme. It has been demonstrated that even a single site mutation in Cu/Zn-SOD is transduced in more flexible regions of the proteins, i.e. the loops surrounding the active site, generating easier accessibility of the copper atom to the substrate, and hence changing the enzymatic rate of reaction [23].

A 'non-specific' way to modulate electrostatic environment is through an external physical stimulus, such as pH or temperature, which, inducing a local or global unfolding of the protein, modifies protein secondary and tertiary structures, thus affecting the rate of reaction. For the SOD1, thermal or pH variations have proven by either experimental or numerical recent results [24–26] to couple with the protein structure having an ultimate effect on its functionality.

Another way to affect the protein electrostatic environment is through the action of electric fields. A significant fraction of cytosolic proteins is exposed to strong endogenous electric fields [27], which have been found capable to modulate enzyme catalysis as demonstrated in experiments utilizing the vibrational Stark effect [28]. For protein crystals the use of an external electric field stimulation in the order of $10^8$ V/m combined with X-ray crystallography has been recently proven to reveal protein mechanics [29], mediated by the coupling of the field with the electrostatics of the protein. Moreover, external electric fields of almost comparable

intensities, exploited in the last years for new technologies at the basis of potential biomedical applications, seem to couple with internal compartments of the cell and among them with proteins. Nanosecond pulsed electric fields (nsPEF) have been proven to activate membrane proteins in response to a single or a repeated pulse application [30], to interact with inner cell organelles [31, 32] and with cytosol proteins and enzymes [33–35].

Despite the wide range of endogenous and/or exogenous electric fields, to our knowledge there is limited evidence of a coupling with SOD1 enzyme. Therefore, given the sensitivity of the SOD1 active site to external physical agents and the widespread use of intense electric fields for cell manipulation, it seemed of particular interest to elucidate how nsPEFs can affect electrostatic environment even in view of the master key role of SOD1 in controlling ROS levels inside cells.

Electrostatic environment of protein active site is usually evaluated by means of computational models. Few experiments, providing evidence of a direct measure of the electrostatic fields at the active site of enzymes through vibrational Stark effect (VSE) applied to specific probes, are available [36, 37]. However, this kind of technique is highly complex [38] and several criteria must be met for calibrated probes transitions to be useful in measuring electric fields inside a protein environment. For example: the probe transition must be incorporated into the protein of interest; its spectrum should be easily separated from that of the protein itself; the labelling of the protein with a non-native bulky dye has the potential to perturb the native electrostatics to such an extent that the biological function under investigation can be significantly altered or destroyed [38].

Conversely, computational models are becoming increasingly affordable [39]. The high performances of computing resources, accompanied by refined protein design methodologies, has allowed the design of increasingly sophisticated proteins with diverse topologies and functions. Protein dynamics ranging from local fluctuations around equilibrium conformations to large-scale conformational changes upon binding can be captured by molecular dynamics (MD) simulations, which use a physics-based potential energy function to simulate protein dynamics as a function of time according to classical Newtonian mechanics. While biophysical techniques, such as NMR spectroscopy, can yield insights into protein dynamics, MD has the power not only to identify functionally relevant conformations, which may be 'hidden' to experimental techniques, but it can also provide the details of transitions between these conformations [39].

The hypothesis we tested using MD simulations, was the possibility to modulate the SOD1 active site electric environment by external intense electric fields similar to the one used in nsPEFs applications.

For these reasons, we observed the response of SOD1 to the application of a 100 nsPEF with different intensities (in line with literature numerical studies from $10^8$ to $7 \times 10^8$ V/m) and different shapes (Monopolar and Bipolar). For electric field intensities below the unfolding threshold, we studied the pattern of the electrostatic distribution at the reaction site of the enzyme, as a marker. We numerically evaluated maps of the electrostatic field distribution of the SOD1 active site, looking for a modulation of the electrostatic environment. Such interaction can be considered as the first step of a cascade of events culminating in overall cell response.

## Materials and methods

### Protein structure

To perform our simulations, we used the structure as reported in [40], where for the purpose of modelling the enzyme core, an available structure of bovine Cu/Zn-SOD complexed with

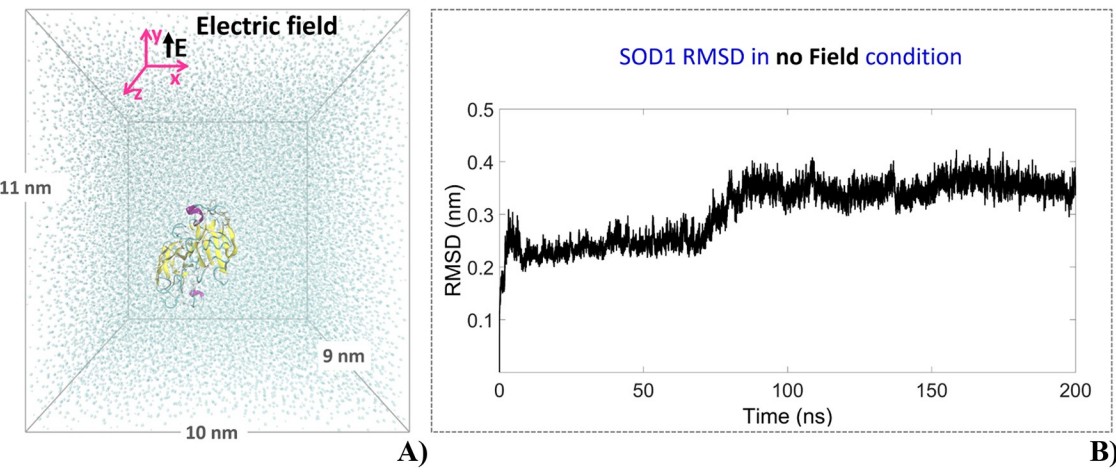

**Fig 1. SOD1 molecular model. (A)** SOD,Cu-Zn molecular model showing the simulation box (10 x 11 x 9 nm$^3$) containing 32292 water molecules, the SOD1 enzyme and 9 Na$^+$ counterions. The SOD,Cu-Zn enzyme is formed by two monomers, each one containing a reaction centre. The electric field is applied in the y direction; **(B)** RMSD of the SOD1 enzyme during equilibration of the 200 ns 'no field' production run.

an azido group (PDB code: 1SXZ) has been equilibrated and minimized replacing the azido group with $O_2^-$ [41, 42].

The environment where the SOD1 enzyme was immersed consists of a rectangular box (10 x 11 x 9 nm$^3$) containing 32292 single point charge (SPC) water molecules with 9 Na$^+$ counterions, for a total of 99659 atoms (Fig 1A). The final system density was 1000 kg/m$^3$. The Cu/Zn-SOD enzyme accomplishes its role through the active site formed by the $Zn^{2+}$ ion, the $Cu^{2+}$ and the superoxide anion $O_2^-$ (Fig 2A).

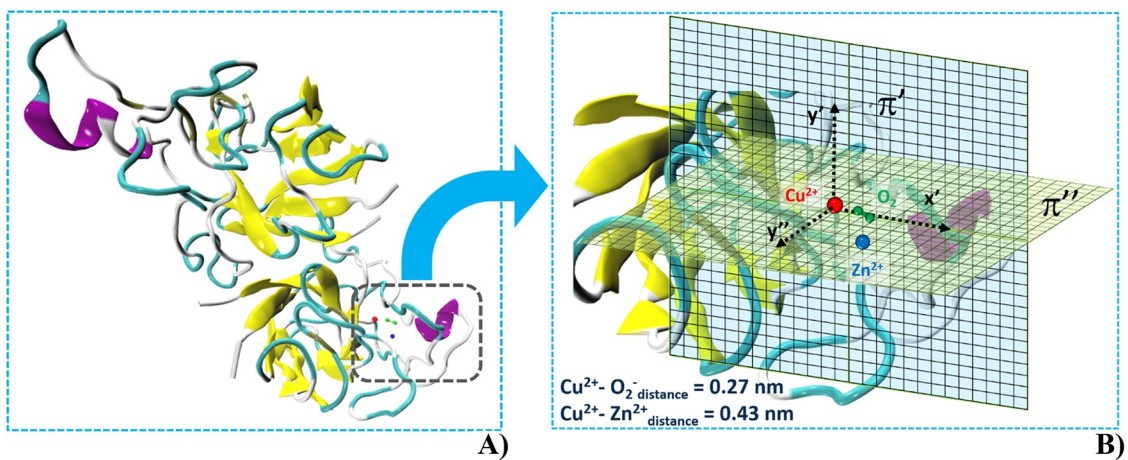

**Fig 2. SOD,Cu-Zn dimer and active site (2D) grid for the *local* E field distribution calculation. (A)** Molecular model of the SOD1 dimer, represented via VMD software [54]. **(B)** Two orthogonal grids with a 4.7 x 4.7 nm$^2$ surface are reported. The π"grid represents a plane passing through the atomic coordinates of $Cu^{2+}$, $O_2^-$, $Zn^{2+}$ with the x'-axis oriented on ($Cu^{2+}$- $O_2^-$) direction and y"-axis normal to it. The plane π' is orthogonal to π"with the x'-axis oriented on ($Cu^{2+}$- $O_2^-$) direction and y'-axis orthogonal to both x' and y", respectively. Some relevant residues within the active site are depicted in the figure, in particular the GLU133, GLU132, ASN131 reported as a purple α-helix and white coil for the glutamatic acid and the aspargine, respectively.

## Molecular dynamics simulations

Following an energy minimization and subsequent solvent relaxation, the system was gradually heated from 50 K to 300 K using short (typically 60 ps) MD simulations using Gromacs package [43]. A first trajectory was propagated up to 200 ns in the NVT (number of particles, volume and temperature are constants) ensemble using an integration step of 2 fs. The temperature was kept constant at 300 K by the V-rescale thermostat [44] which provides a consistent statistical mechanical behavior. All bond lengths were constrained using the LINCS algorithm [45]. Long range electrostatics were computed by the Particle Mesh Ewald method [46] with 34 wave vectors in each dimension and a 4th order cubic interpolation. The GROMOS96 force field [47] parameters were adopted. Short range interactions were evaluated within a 1.1 nm cut off radius.

Once obtained an extended equilibrated-unexposed trajectory (Fig 1B) we evaluated possible effects due to the intensity and the specificity of a single nsPEF by applying either a Monopolar or a Bipolar electric pulse (referred to hereinafter as MP and BP, respectively). The MP presents rise and fall times of 2 ns duration and a hold time ($t_{ON}$) of 100 ns; the trajectory has been prolonged for 50 ns ($t_{OFF}$) after the pulse switch-off. The BP presents the same rise and fall times durations, while a 50 ns duration for the positive pole ($t_{ON\ positive}$), before reverting the amplitude of the field for the following 50 ns ($t_{ON\ negative}$). The trajectory has been prolonged for 50 ns ($t_{OFF}$) even in this case.

We implemented these MP and BP signals inside MD simulations by modifying the *sim_util* library inside the Gromacs package. The E field intensity for both signals was ranged from $10^8$ to $7x10^8$ V/m, acting on all atoms within the simulation box as explained in [48].

## Secondary structure analysis

To evaluate possible structural changes induced by the external signals considered, we calculated the number of secondary structures through the DSSP program [49]. The calculation has been made in terms of the mean value of β-Sheet and Coil number during the on phase of both MP and BP, with respect to the equilibrium condition (no E field applied).

## RMSD, radius of gyration and solvent accessible area

To quantify the protein atoms positions during the MD simulation we computed the RMSD (root-mean-square deviation). The RMSD provides a quantitative measure of the protein structural variations, comparing the electric field exposure and the equilibrium state [43]. To quantify the distribution of the atoms in the space relative to their own centre of mass, we also computed the radius of gyration, which can provide an understanding of the changes in shape and size of the protein under the influence of external stress. The extent to which an amino acid interacts with the solvent and the protein core is naturally proportional to the surface area exposed to these environments, hence we calculated both the hydrophobic and hydrophilic solvent accessible surface area (SASA) [43].

## Dipole moment spectrogram

Usually proteins, due to their secondary structure conformation (α-helices, β-sheets, turns, coils, etc.) possess an electric dipole moment and when an external electric field is applied, the protein orients itself in the direction of the field. The spectrogram is a time-frequency representation, able to describe the spectral content of a signal, obtained as the square modulus of the short time Fourier transform algorithm. This algorithm is implying the windowing of the temporal signal under analysis and the fast Fourier transform of each time sequence. Here,

temporal sequences $4.0 \times 10^{-11}$ s long filtered through a Hamming passband filter [50], with 50% overlap between adjacent segments have been chosen. We obtained a frequency resolution of 0.25 GHz and a time resolution of 6 ns.

In our study, the electric field was applied in the y-axis (Fig 1A). We valued these changes by the analysis of the frequency spectral content of the dipole moment of protein and water molecules. This outcome has been evaluated both considering the application of an electric field of $10^8$ V/m and a temperature increase of 35 degrees (335 K, SOD1 melting temperature [51]) to compare the different molecules' response.

## Electrostatic distribution at the active site

With the aim to study the rigorous electric field distribution within the active site region, we constructed local electric field maps (see Eq (1)), given by the explicit coulombic contribution of all surrounding atoms (protein, water and Na ions), as already performed for another protein in [52]. This is in line to what noted in [53] where it is suggested that implicit models based on Poisson function and values of dielectric constants can give an inadequate description of the heterogeneous protein environment.

We investigated the effects of a single MP or BP 100 nsPEF with an intensity of $E_y = 10^8$ V/m, comparing the results with the ones obtained in equilibrium conditions. We performed such calculations on two representative 2D planes (4.7 x 4.7 $nm^2$) centered in the $Cu^{2+}$ position (see Fig 2). The first one ($\pi''$) passing through the coordinates of $Cu^{2+}$, $O^{2-}$ and $Zn^{2+}$ (with the x'-axis oriented on ($Cu^{2+}$- $O^{2-}$) direction and y''-axis normal to it); the second one ($\pi'$) orthogonal to $\pi''$ (with the x'-axis oriented on ($Cu^{2+}$- $O^{2-}$) direction and y'-axis orthogonal to both x' and y''). Such planes (Fig 2B) have been densely meshed (with a spatial resolution around 0.05 nm), resulting in a grid with several nodes, where the electric field results:

$$E_{r_i} = \sum_n \frac{q_{n,protein}}{4\pi\varepsilon_0 |(r_{n,protein} - r_i)|^3} (r_{n,protein} - r_i) + \sum_n \frac{q_{n,water}}{4\pi\varepsilon_0 |(r_{n,water} - r_i)|^3} (r_{n,water} - r_i) +$$
$$+ \sum_n \frac{q_{n,ion}}{4\pi\varepsilon_0 |(r_{n,ion} - r_i)|^3} (r_{n,ion} - r_i) \tag{1}$$

where $r_i$ stands for node's position and $r_{n,species}$ indicates the n-th atom position with charge $q_{n,species}$ belonging to each chemical species involved in the production of the perturbing field. Eq (1) is thus made of three terms: the first one representing the perturbation due to each aminoacidic residue of the protein, the second which considers the perturbation due to local electric field generated by water molecules, the last one due the presence of counterions.

Finally, we focussed on specific charged residues (i.e. glutamic acids, Fig 2A purple colour) surroundings the SOD1 active site to envisage a possible electrostatic guidance of the $O_2^-$ ion to interact with the copper $Cu^{2+}$ mediated by the external electric field application.

## pKa calculations

pKa values of specific ionisable protein residues have been obtained via the PROPKA algorithm [55]. The software predicts the pKa values of ionizable groups in proteins and protein-ligand complexes based on the 3D structure. The algorithm permits fast and accurate determination of the protonation states of key residues and ligand functional groups within the binding or active site of a protein.

## Statistics

The variations of secondary strucures and pKa values are reported as Mean±S.D. The statistical differences were analyzed trough the KaleidaGraph program (Synergy Software, Reading PA, USA) by applying the two-sided Students t-test. P values <0.05 were considered statistically significant and indicated as follows: *P<0.05; **P<0.01; ***P<0.005.

## Results

### Electric field threshold for structural effects

A comparative analysis is carried out between the structure of SOD1 enzyme in "No- field" conditions and when exposed to the external electric field stimulus. The goal is to define a threshold for the intensity of the E field able to electrically or geometrically manipulate the SOD1 enzyme.

To this end, the average numbers of Coil and β-Sheet secondary structures have been evaluated for decreasing field intensities (see Fig 3, panel C), comparing the "No field" condition with a set of MP and BP signals. Student's t test revealed a threshold for relevant conformational changes (above 10%) for an E field intensity of $5 \times 10^8$ V/m. The highest intensity used ($7 \times 10^8$ V/m) produces dramatic structural rearrangements, probably associated to an

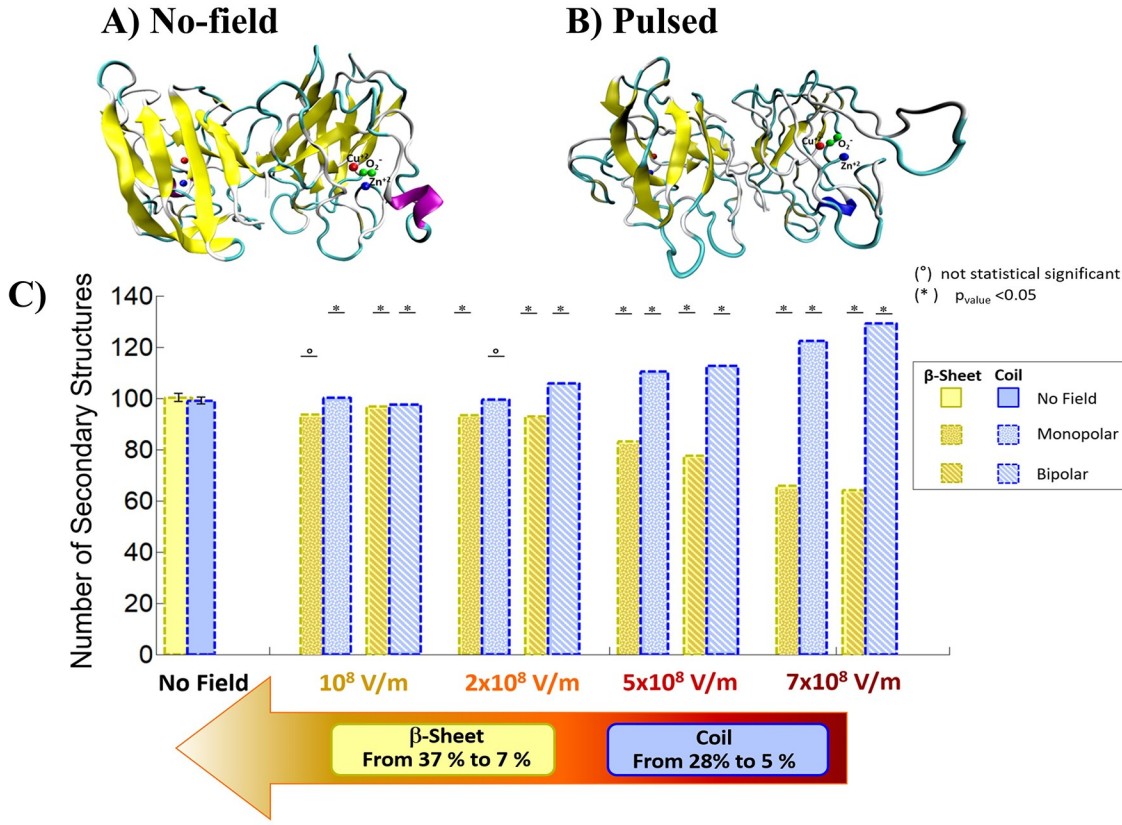

**Fig 3. SOD1 secondary structures analysis.** Panel A and B. Comparison of the conformation of SOD1 before exposure (A) and after an exposure to a Bipolar pulse with $7 \times 10^8$ V/m intensity (B). Panel C. Coil and β-Sheet secondary structures mean values. The results are presented in no exposure condition and under MP and BP 100 nsPEF with intensities ranging from $10^8$ to $7 \times 10^8$ V/m. The asterisk (*) indicates p value smaller than 0.05 (p<0.05) which means a variation statistically significant, and the circle (°) indicates no significant differences. The statistical analysis has been performed with the Student t test. Bars on graph indicate the standard error.

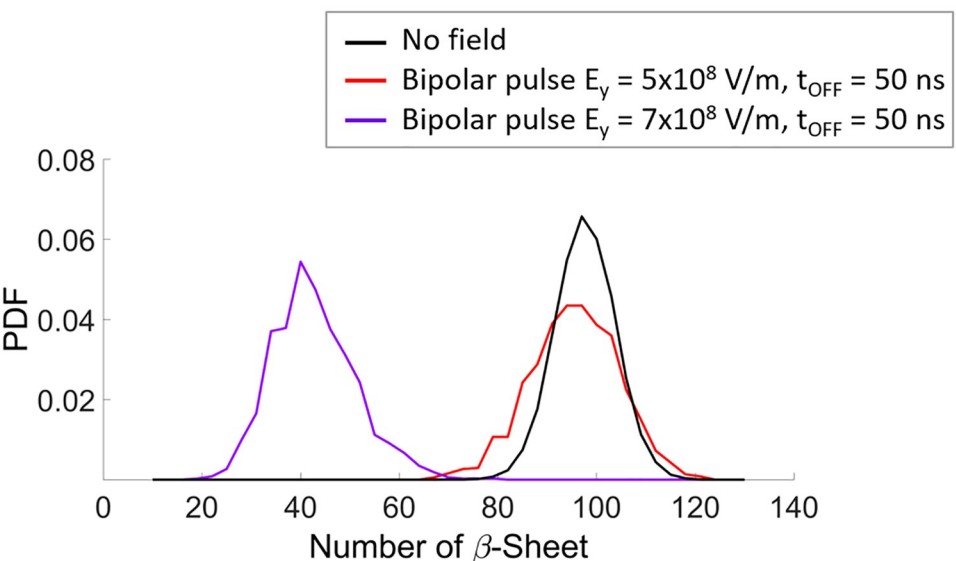

**Fig 4. SOD1 secondary structures recovery.** Probability density function of SOD1 β-Sheets during the OFF state (50 ns) after the BP ($5x10^8$ and $7x10^8$ V/m) switch off.

irreversible unfolding transition [56], since we observed a maximum loss of β-Sheets up to 37% and an increase of the coil secondary complexes up to 28%. For the lowest field intensities adopted ($10^8$ and $2x10^8$ V/m) the effects are subtle ($< 5\%$ structural variation either for the coil structures or for β-Sheets ones) and not statistically significant.

Interestingly, the degree of structural rearrangement occurred during the pulse application, drives the protein unfolding transition to an 'irreversible unfolded state' or in a 'recoverable' unfolded state; this is observed following the protein evolution after the E field switch-off. Fig 4 highlights that 50 ns after the $5x10^8$ V/m BP (producing a loss of about 23% in secondary structures, see Fig 3, panel C) switch-off the protein undergoes a full structural recovery. On the contrary, a single $7x10^8$ V/m BP (causing a loss of about 37% in secondary structures, see Fig 3, panel C) produces a significant shift in β-Sheets number with no significant structural recovery even 50 ns after the switch-off, but rather, a further decrease in β-Sheets number up to 55% (see the violet distribution in Fig 4).

The fact that a $10^8$ V/m electric pulse is not able to affect SOD1 geometry can also be inferred by the analysis of a set of standard biophysical observables, i.e. the RMSD, the radius of gyration and the SASA (both hydrophobic and hydrophilic solvent accessible surface areas). Illustrative results are reported in Fig 5 for the No–Field condition and for the $10^8$ V/m MP.

In Fig 5A the probability density function (PDF) of the RMSD is presented, showing for the "No-field" condition a bimodal distribution with peaks around 0.2 and 0.35 nm, indicating that the system is fully equilibrated as also reported in Fig 2 [40]. Additional insights into the protein changes in terms of shape and size under the influence of external stresses can be inferred by the radius of gyration (Fig 5B). The data for the No-field condition shows a quite packed enzyme with the root mean square distances of the protein's atoms from its center of mass around 1.9 nm, with fluctuations within 0.1 nm during the whole trajectory. Finally, the total solvent accessible area, split as the PDF of the hydrophobic and hydrophilic areas, respectively, are reported in Fig 5C and 5D.

The $10^8$ V/m MP is not able to induce noticeable changes in any of the observables considered, confirming that such intensity is not able neither to induce structural modifications of

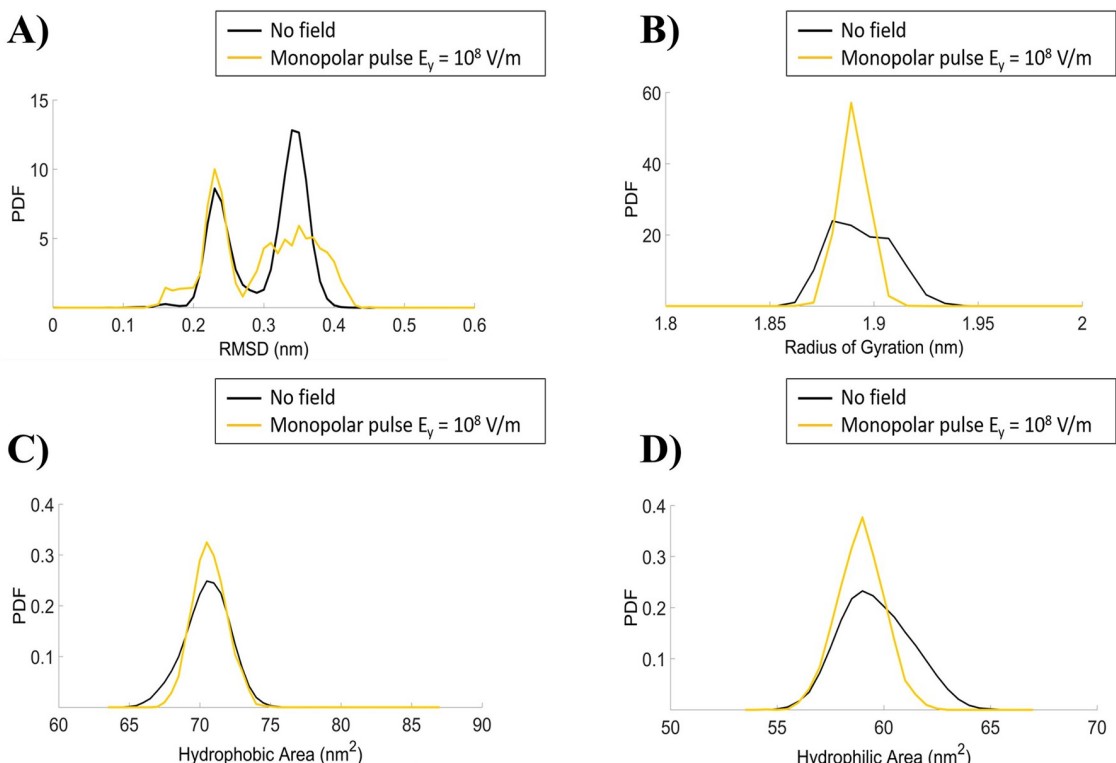

**Fig 5. MD simulations observables.** In the panel four different protein observables are presented as probability density distributions: the RMSD (A), the Radius of gyration (B) and the Hydrophobic (C) and Hydrophilic (D) areas. The curves refer to the protein structure in equilibrium condition (black line) and under the effect of external MP of $10^8$ V/m intensity (yellow line).

the protein nor to determine any variation in the parameters usually adopted for monitoring protein changes. Similar results have been obtained for the BP (see S1 Fig).

### Electrostatic coupling mechanism

Although the lowest intensity of $10^8$ V/m is not able to induce structural changes, a not negligible coupling with the protein electrostatics is expected [48], due to the dipolar mechanism, either mediated by the solvating water or acting straight on the enzyme.

Under the influence of an external E field, the protein residues, as well as water molecules, are subject to a reorientation quantified by their dipole moment.

To this end, we calculated both the protein and the solvation-water dipoles alignment under MP and BP signals, comparing them to the values obtained without any exposure. In Fig 6 such a coupling is evaluated by means of the spectrogram of the dipole y-component $M_y$ (spectrograms associated to the total dipole ($M_{TOT}$) are reported in S2 Fig for a complete view), since the polarization process essentially occurs along the exogenous field direction (see Fig 1). A spectrogram is a visual representation of the spectrum of a signal as it varies with time and it is built from a sequence of spectra stacking them together in time and by compressing the amplitude axis into a 'colour map'.

In No-field condition, (Fig 6, 1st row, 2nd column), the frequency content associated to water dipole dynamics easily reaches the 1 GHz limit for the whole observation time, while a lower frequency of about 700 MHz, or less, is observed for the protein dipole dynamics (see Fig 6, 1st row,1st column). When the $10^8$ V/m MP or BP are applied, we witness a strong

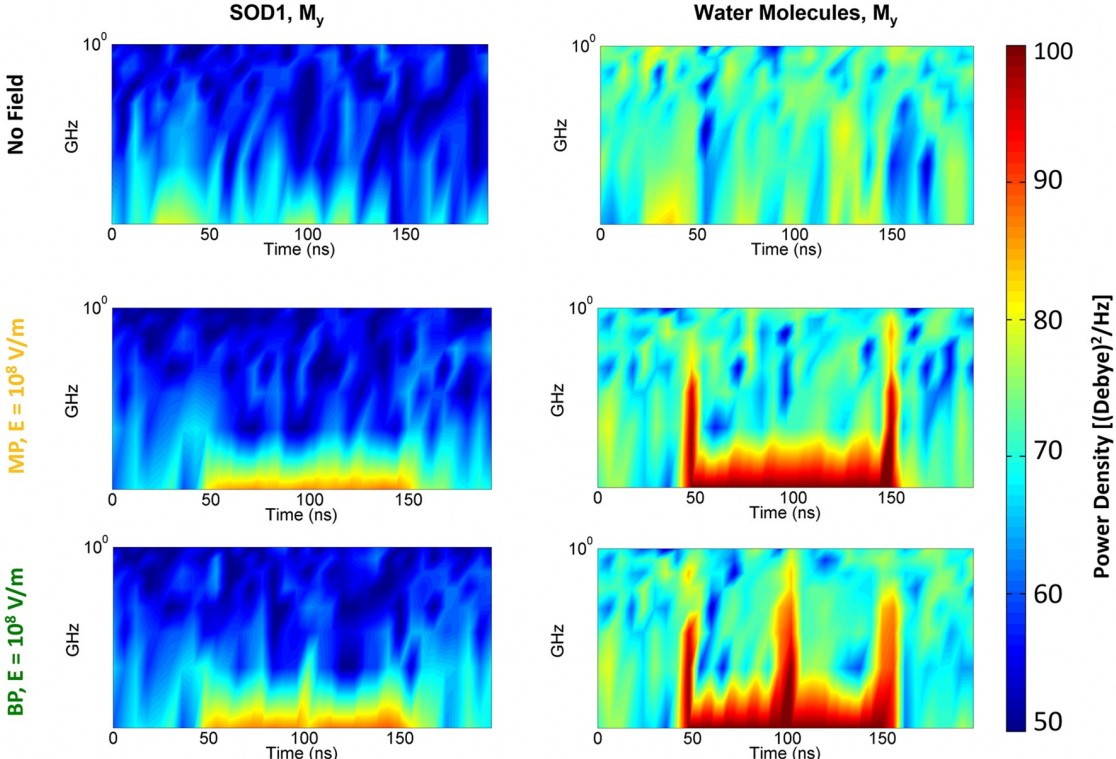

**Fig 6. Frequency spectral content of the SOD1 and water dipole moment.** Spectrogram representation of the dipole moment (y-component) of both Cu,ZnSOD1 protein and solvating water medium, reported in absence of any exogenous field (black label, first raw) and in presence of a 100 ns, $10^8$ V/m MP (yellow label, second raw) and BP (green label, third raw).

dipolar reorientation of water molecules, which amplifies the spectrogram response in correspondence of the signal variations (note the output spectral power density during the signal's rise and fall times). In particular, for the MP (Fig 6, 2nd raw), the rise and fall times of the signal are clearly visible, coupled to an increase in the output spectral power density. Similarly, for the BP (Fig 6, 3rd raw) the reversal of the signal polarity is clearly observable in the increase of the spectral content at the middle of the signal course.

As expected, when considering the behaviour of the protein dipole alone a similar trend is observed, although with marked lower spectral power densities. This is essentially due to the relatively low polarizability of the target. Therefore, SOD1 needs more time to reorient, nonetheless it is still able to couple to the external electric signals.

## Electrostatic coupling on SOD1 active site

To fully exploit the potentialities of the external electric field to affect the electrostatic environment of SOD1, we investigated the *local* electric field distribution on the active site of one of the two dimers (dashed in Fig 2A and magnified in Fig 2B).

Our analysis was driven by previous works by D'Alessandro and co-workers [40], [42], where authors, combining mixed QM/MM methods with basic statistical mechanical relations, were able to study the chemical events and the atomic motions of the complex environment of the SOD1 reaction centre, pointing out a dramatic effect of the electrostatic field due to protein and solvent interactions on free energy surface at the active site.

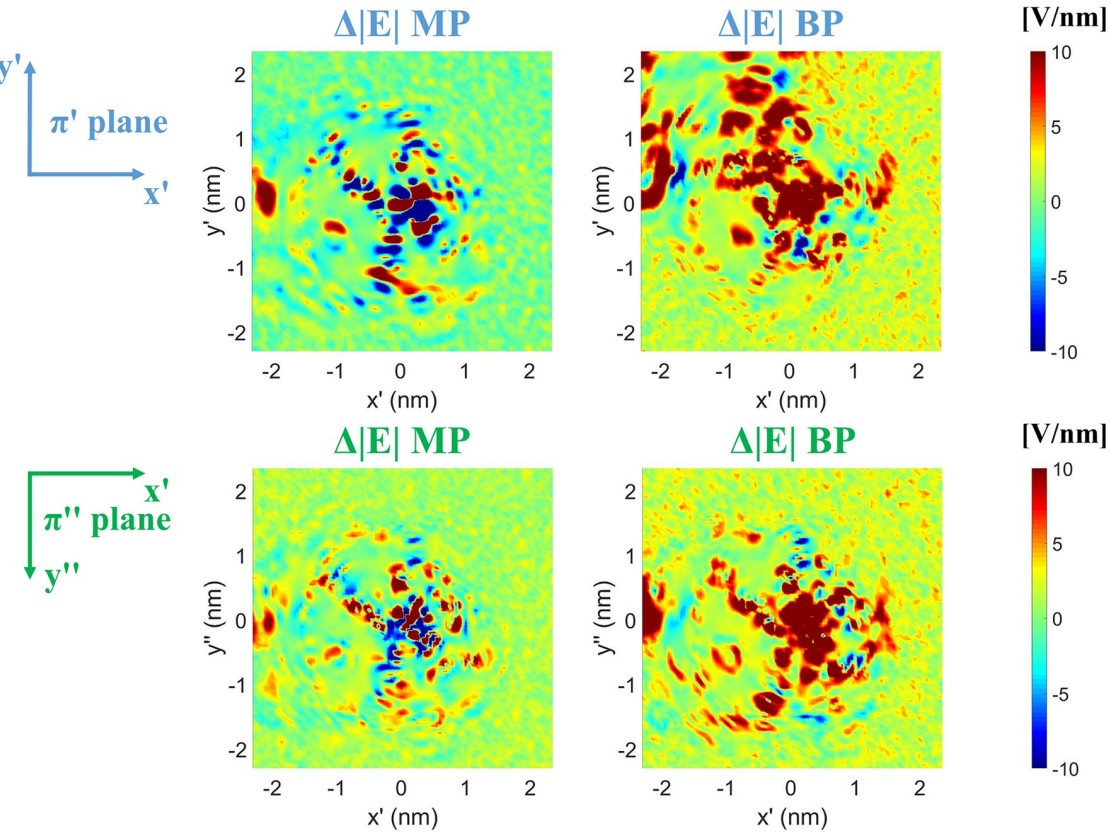

**Fig 7. 2D-maps of the *local* electrostatic field modification around the active site.** Electric field effects on the π' and π" planes, due to both MP (first column) and BP (second column) signals, are shown.

Here we went to a finer evaluation of the contribution of protein and solvent to the active site in terms of maps of electric field distribution with a sub-Angstrom spatial resolution. We show in Fig 7 the electric field shifts due to the applied MP and BP signals on the two representative planes π' and π" described in Materials and Methods (see S3 Fig for the electric field map in the reference no-field condition). Both signals are able to modify the electrostatic map of the active site in the central region around the copper ion position. However, the change produced by the BP presents a more diffused and uniform distribution with the same sign, suggesting a possible rotation of specific protein residues, and hence a charge distribution alteration, due to the external field action.

In order to have a further indication of the external field effect on the active site environment, following some interesting studies claiming that local electrostatic fields can perturb pKa values of ionisable residues [57], we calculated pKa values for the four residues: His 63 and His 120, Asp 83 and His 71, known to be fundamental for the reaction process, since they affect the electron transfer process as reported in [40].

In Fig 8 we present pKa data of the four residues for the BP pulse, distinguishing two contributions: during the pulse and the 50 ns after the pulse itself. Results indicate a clear shift of pKa values, suggesting that the $10^8$ V/m BP signal is able to affect the local environment at SOD1 reactive site. In particular the most significant data (p<0.001) are observable after the pulse, indicating a cumulative effect on all the residues. Moreover, the most dramatically affected seems to be the Asp 81, where the effect is related to an inversion of the sign of pKa with respect to the no-field condition.

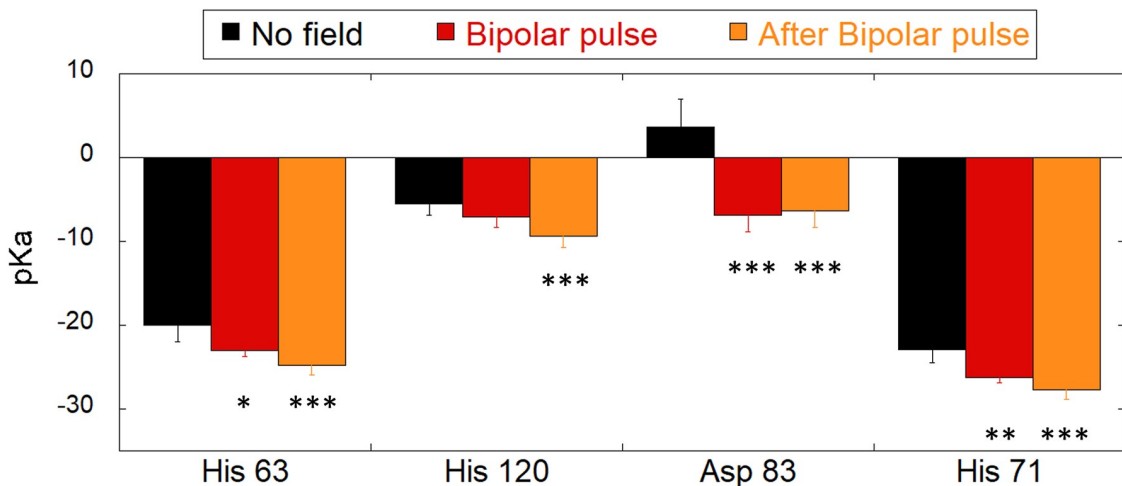

**Fig 8. pKa values shifts of four specific protein residues.** pKa values of His 63, His 120, Asp 83 and His 71 calculated before BP application (black hystograms), during the negative pole of the BP and after BP removal. Student-t test has been applied to verify the statistical significance of pKa negative shifts in presence and after the BP application.

Negative variations of pKa values are associated to the build-up of a negative charge on residue side chain. Interestingly, such effects are consistent with the ones presented in Fig 7, where the electric field shifts due to the applied BP pulse were supposed to be a consequence of possible rotation and charge distribution alteration of specific protein residues.

## Discussion

In this paper we chose to explore the effects of MP and BP nsPEFs coupling with the electrostatics of the Cu/Zn-SOD enzyme. The aim has been to define the intensity of the field able to interact with the SOD1 enzyme and to modulate the electrostatic environment at the active site without implying major changes in protein conformation. Besides this, thresholds for reversible and irreversible denaturation of the SOD1 under the application of nsPEFs have been identified. MP and BP signals, chosen as representative of typical nsPEFs signals, have been implemented for the first time in the Gromacs environment for MD simulations.

The coupling with the applied external electric field can be split in a direct action of the field on the dipolar aminoacidic residues of the enzyme and an indirect action of the field mediated by the presence of the solvating water. It has already demonstrated in the past how solvating water can interact with intense electric fields [58–60]. By the analysis of the coil and β-Sheets secondary structures, we evidence here the direct contribution of the field. We identified three different electric field thresholds for structural effects: *(i)* with E field intensities lower than $2\times10^8$ V/m small (and fully reversible) changes (5%) were appreciated; *(ii)* when a E field of $5\times10^8$ V/m is applied, variations around 23% are reported, with an almost complete structural recovery within 50 ns after the E pulse removal; *(iii)* a severe denaturation when the protein is exposed to an electric pulse of $7\times10^8$ V/m, not reversible within the 50 ns switch-off period. For completeness, also a thermal perturbation has been considered, in order to compare the structural effects of a 35˚K (see S4 Fig) temperature increase, which is known to experimentally induce a temperature-dependent unfolding transition [51], with the ones induced by the field. The simulation data, obtained by heating the system, show no effects, conceivably due to the different kinetics of temperature-induced and E field-induced unfolding transitions. This also confirms what already predicted in literature [56], where pulsed

electric fields induced structural changes in the secondary structure of lysozyme that are not equivalent to those caused by thermal stress.

The electric field intensities identified are in line with modelling and experimental literature, ranging from $10^7$ to $7\times10^8$ V/m, since it is already known that intense electric fields are needed to act on biomolecular processes inside cells. The majority of membrane proteins are naturally exposed to strong electric fields that range in strength from $10^4$ to $10^7$ V/m [61] and the ability of electric fields to modulate the structure of integral membrane proteins is a central dogma in voltage gating [62]. Conversely, cytosol proteins are less studied in presence of E fields. To this regard, apart from few studies that have an experimental validation [56], the majority of the investigations are realized on the basis of molecular simulations. In particular, several endpoints have been explored using MD simulations [63–71] such as: the effects of external electric field on the stability of protein conformations [64–66], the stability of β-Sheets structures [69–71], the dependence of the field on the polarization of water molecules and the role of water in hydrophilic proteins, where in the interfacial gap, it forms an adhesive hydrogen-bond network between the interfaces, stabilizing early intermediates before native contacts are formed [72]. Almost all the papers reporting effects on the coupling of electric fields and proteins refer to intensities higher than $10^8$ V/m [73–77], even in the case of recent papers giving contributions in comprehending the complete unfolding process of a protein [78].

Besides the identification of the unfolding pathways through the analysis of secondary structures, we also quantified the SOD1 protein structural rearrangements through the computation of the RMSD, the radius of gyration and the solvent accessible area. For the lowest intensity here used ($10^8$ V/m) negligible effects are observed.

To explore the polarization of the system we studied the spectrograms of both the protein and the solvating water dipoles, comparing the reference (No field) condition with the ones relative to the two kinds of electric pulses considered ($10^8$ V/m MP and BP). When the $10^8$ V/m electric stimulus is applied, the SOD1 dipole follows the external signal in its shape, showing a high degree of polarizability when compared to the reference condition. This result suggests an efficient electric coupling between the external E field and the SOD1 protein.

In the study previously cited [21], the electrostatic guidance mechanism has been computationally exploited to enhance a reaction (i.e. oxygen and hydrogen peroxide at the active-site $Cu^{2+}$ ion) that is rate-limited by diffusion. Getzoff and colleagues have shown that the site-specific mutants produce electrostatically facilitated diffusion rates, maintaining the detailed structural integrity of the active-site electrostatic network.

In the present paper, we envisage a critical role of the electric coupling between the protein dipole moment and the MP and BP signals in modifying the electric forces acting on the Cu/Zn-SOD active site. Strengthened by use of computational methods based on MD, we have been able to carry out this study in terms of highly resolved 2D maps of the electric field distribution on two planes passing through the $Cu^{2+}$ and $O_2^-$ coordinates. By the investigation of the electric field maps, both MP and BP produced significant electrostatic shifts with respect with the reference condition. In particular, a broader square area spanning 1 nm apart in both directions of the plane from the copper ion position, is visible for both MP and BP, but far more relevant for the BP. Moreover, relevant pKa changes in four important residues (among the ones involved in the active site) suggest possible effect on the local environment at SOD1 reaction centre.

The question is: can we speculate about possible consequences of an E field-induced perturbation at the active site on the diffusion of $O_2^-$ radical towards the $Cu^{2+}$? If one looks at the electric field shift in the central region of both the representative planes π' and π", where the radical approaches the copper ion prior to the electron transfer process [40] (see Fig 2b), islands of positive electric field gradients emerge, to the point that the corresponding forces

can act on $O_2^-$ ion to drive its movement in the desired direction (the BP, in particular, has proved to provide excellent results). This means that sufficiently high intense electric pulses could, in principle, establish a sort of *ion-trap* in a distance-dependent fashion [79]. Even in this case, the thermal stimulus does not affect the electrostatic environment hereby described (see S4 Fig).

## Conclusion

In conclusion, the investigation here reported unveiled different modalities for the interaction mechanisms between ultra-short pulsed electric fields and the enzyme target.

1. The lowest field used, $10^8$ V/m, turned out to be a threshold intensity able to affect the electrostatic environment of the enzyme active site. Such electrostatic variation may as a final step act as an effective electrostatic guidance. This happens without irreversible modifications or impact on protein functions, as derived from the dipole moment analysis, showing a protein that can reorient itself following the external field applied at the threshold intensity of $10^8$ V/m without irreversible polarization (i.e. after the signal's switch-off the structure fully recovers its physiological polarization state).

2. Significant structural changes are visible when electric field intensities starting from $5 \times 10^8$ V/m are applied. Interestingly, two different protein fates are predicted: i) the 100 ns $5 \times 10^8$ V/m MP and BP signals are not able to induce a complete unfolding transition for the SOD enzyme, with a partially denatured state (obtained at the end of the 100 ns pulse) fully recovered during the signal's switch-off period; ii) the 100 ns $7 \times 10^8$ V/m MP and BP produced a fast transition (occurring within few ns) between folded and unfolded states, as inferred by secondary structures and geometrical analysis. In this case, the signal's switch off does not produce any significant structural recovery.

3. The signal's characteristics seem another key-point in the interaction mechanism, as apparent from Figs 3, 7 and 8. In the present study, the BP signal turned out to produce a more efficient coupling with the enzyme, both in terms of structural and dipolar effects.

This numerical study can be considered as a starting point for possible prediction of future experiments on the superoxide dismutase protein and it can be considered as the basis to investigate the interaction of nsPEFs electric fields with enzymes.

## Supporting information

**S1 Fig. MD simulations observables for the BP signal.** In the panel four different protein observables are presented as probability density distributions: the RMSD (A), the Radius of gyration (B) and the Hydrophobic (C) and Hydrophilic (D) areas. The curves refer to the protein structure in equilibrium condition (black line) and under the effect of external MP of $10^8$ V/m intensity (light blue line).
(TIF)

**S2 Fig. Frequency spectral content of the whole simulation system.** Spectrogram representation of the dipole moment (y-component) of all chemical species inside the simulation box, reported in absence of any exogenous field (black label) and in presence of a 100 ns, $10^8$ V/m MP (yellow label) and BP (green label).
(TIF)

**S3 Fig. 2D-maps of the local electrostatic field in No field condition.** 2D-maps of the *local* electric field (absolute value depicted) around the active site on the π' plane and the π" plane in

the No field condition (first and second raw respectively).
(TIF)

**S4 Fig. Effect of a temperature increase on the frequency spectral content of the total dipole moment.** Effect of a 35K temperature increase on the frequency spectral content of the dipole moment (y-component) of all chemical species inside the simulation box (first column), the Cu,ZnSOD1 alone (second column) and the water molecules (third column).
(TIF)

## Acknowledgments

Authors want to thank Andrea Amadei and Massimiliano Aschi for their support in the theoretical interpretation of the target enzyme molecular modelling and for their support in approaching molecular dynamics simulations.

## Author Contributions

**Conceptualization:** Paolo Marracino, Micaela Liberti, Francesca Apollonio.

**Data curation:** Elena della Valle.

**Formal analysis:** Elena della Valle, Paolo Marracino.

**Methodology:** Paolo Marracino.

**Project administration:** Francesca Apollonio.

**Supervision:** Micaela Liberti, Francesca Apollonio.

**Validation:** Elena della Valle, Olga Pakhomova.

**Writing – original draft:** Elena della Valle.

**Writing – review & editing:** Paolo Marracino, Olga Pakhomova, Micaela Liberti, Francesca Apollonio.

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
