## [Decision Letter · Decision Letter 0]

28 Jun 2019

PONE-D-19-16769

Nanosecond pulsed electric signals can affect electrostatic environment of proteins below the threshold of conformational effects: the case study of SOD1 with a molecular simulation study

PLOS ONE

Dear Dr. Apolonio,

Thank you for submitting your manuscript to PLOS ONE. After careful consideration, we feel that it has merit but does not fully meet PLOS ONE’s publication criteria as it currently stands. Therefore, we invite you to submit a revised version of the manuscript that addresses the points raised during the review process.

ACADEMIC EDITOR: Please try to improve your manuscript according to the serious reviewers' criticism.

We would appreciate receiving your revised manuscript by Aug 12 2019 11:59PM. To enhance the reproducibility of your results, we recommend that if applicable you deposit your laboratory protocols in protocols.io, where a protocol can be assigned its own identifier (DOI) such that it can be cited independently in the future. For instructions see: http://journals.plos.org/plosone/s/submission-guidelines#loc-laboratory-protocols

We look forward to receiving your revised manuscript.

Kind regards,

Eugene A. Permyakov, Ph.D., Dr.Sci.

Academic Editor

PLOS ONE

Journal Requirements:

1.When submitting your revision, we need you to address these additional requirements.

2. Thank you for stating the following in the Financial Disclosure  section: "The author(s) received no specific funding for this work."

We note that one or more of the authors are employed by a commercial company: Rise Technology srl.

Reviewers' comments:

Reviewer's Responses to Questions

**Comments to the Author**

1. Is the manuscript technically sound, and do the data support the conclusions?

Reviewer #1: Yes

Reviewer #2: Partly

2. Has the statistical analysis been performed appropriately and rigorously? 

Reviewer #1: Yes

Reviewer #2: No

3. Have the authors made all data underlying the findings in their manuscript fully available?

Reviewer #1: Yes

Reviewer #2: Yes

4. Is the manuscript presented in an intelligible fashion and written in standard English?

Reviewer #1: Yes

Reviewer #2: Yes

5. Review Comments to the Author

Reviewer #1: The authors try to explore the effect of electric field on enzyme active sites

by exploring the response to field of SOD.

Unfortunately, the paper does not present a comparison between calculated and observed values. This includes the argument that the rate changes.

Unless I missed something the paper is compared to some experiments

I cannot see advantage in publication

Reviewer #2: The authors present a molecular dynamics simulation study exploring the effect of electric field pulses on the superoxide dismutase protein. There are two main motivations for this study as presented by the authors: (i) to understand the structural stability of the protein in respect to an external electric field and (ii) to explore the possibility that the external electric field modifies the electrostatic guidance, i.e., the local electric potential that drives the superoxide to Cu2+ for the catalytic electron-transfer reaction. The goal (i) was already studied in a number of previous papers and the authors confirm the basic conclusion that sufficiently strong fields lead to partial protein denaturation. Question (ii) is potentially more novel and I believe here the authors' analysis was insufficient.

The authors explore the distribution of the electric field, and its modification by an external pulse, in the vicinity of the active site. The first question to ask here: why should one care about the electric field? The ability of the superoxide to reach the active site should be driven by the potential of mean force, a free energy, and not by the electric field, which provides the force at a local point. I believe the authors are not calculating the property which is critical for the question they have posed. I also have significant questions regarding how the electric field was calculated. The calculation is based on eq 1, which is the Coulomb law in vacuum. The resulting numbers are based on the magnitudes of partial charges and the distances to them. This is clearly not the entire picture. Electrostatic interactions are screened by water. Near the protein surface one cannot simply used the bulk dielectric constant of 78 and one has to specifically calculate the electric field by the water dipoles. However, the crudest estimate suggests that the numbers presented by the authors are overestimated by a factor of ~78. This is clearly not acceptable. In addition, I assume simulations were done in the standard Ewald protocol. Therefore, Ewald corrections have to be used in the calculation of the electric field as well. The calculations presented by the authors have no physical meaning unless these problems are addressed.

It would be useful to have a consistency check for the electrostatic calculations. Can pKa be calculated to make sure the results are solid? None of the plots presented in the paper are testable by observations. The authors should make some minimum effort to connect to the observable reality.

6. PLOS authors have the option to publish the peer review history of their article (what does this mean?). If published, this will include your full peer review and any attached files.

Reviewer #1: No

Reviewer #2: No

---

## [Author Response · Author response to Decision Letter 0]

11 Aug 2019

Reviewer #1: The authors try to explore the effect of electric field on enzyme active sites by exploring the response to field of SOD.

Unfortunately, the paper does not present a comparison between calculated and observed values. This includes the argument that the rate changes. Unless I missed something the paper is compared to some experiments. I cannot see advantage in publication

It seems that our initial statement was not sufficiently clear, and we apologize for this.

We tried to improve clarity changing the introduction, materials and methods and discussion as evidenced in the text of the new version of the manuscript.

The manuscript is a fully computational study aimed to investigate possible response of the SOD1 enzyme to the application of intense electric field pulses. We do not address specific experimental data to compare with, but the references to experimental works are aimed to demonstrate that a response is feasible and worthwhile to be deepen, due also to the biomedical and biotechnological potentialities as evidenced in [Beebe SJ. Considering effects of nanosecond pulsed electric fields on proteins. Bioelectrochemistry. 2015; 103: 52–59].

Computational models are becoming increasingly affordable [M C Childers and V Daggett, Insights from molecular dynamics simulations for computational protein design, Mol. Syst. Des. Eng., 2017, 2, 9]. The high performances of computing resources, accompanied by refined protein design methodologies, has allowed for the design of increasingly sophisticated proteins with diverse topologies and functions. Protein dynamics, ranging from local fluctuations around equilibrium conformations to large-scale conformational changes upon binding can be captured by molecular dynamics (MD) simulations, which uses a physics-based potential energy function to simulate protein dynamics as a function of time according to classical Newtonian mechanics. While biophysical techniques, such as NMR spectroscopy, can yield insights into protein dynamics, MD has the power not only to identify functionally relevant conformations, which may be ‘hidden’ to experimental techniques, but it can also provide the details of transitions between these conformations [M C Childers and V Daggett, Insights from molecular dynamics simulations for computational protein design, Mol. Syst. Des. Eng., 2017, 2, 9].

The hypothesis we tested using MD simulations, was the possibility to modulate the SOD1 active site electric environment by external intense electric fields as the ones produced in nsPEFs applications. For these reasons, we observed the response of SOD1 to external electric fields, studying whether this enzyme results sensitive to the application of a 100 nsPEF with different characteristics.

Fully atomistic MD studies are acquiring and increasing importance and are finding increasing acceptance in top level journal as PlOS ONE. For example only in the year 2019 there are 8 fully computational (MD) papers puplished in PlOS ONE (see below). Therefore, we believe that even in absence of experimental data our manuscript can be of interest of this journal.

• Daghestani M, Purohit R, Daghestani M, Daghistani M, Warsy A (2019) Molecular dynamic (MD) studies on Gln233Arg (rs1137101) polymorphism of leptin receptor gene and associated variations in the anthropometric and metabolic profiles of Saudi women. PLoS ONE 14(2): e0211381. https://doi.org/10.1371/journal.pone.0211381

• Kandeel M, Kitade Y, Al-Taher A, Al-Nazawi M (2019) The structural basis of unique substrate recognition by Plasmodium thymidylate kinase: Molecular dynamics simulation and inhibitory studies. PLoS ONE 14(2): e0212065. https://doi.org/10.1371/journal.pone.0212065

• Silva SG, da Costa RA, de Oliveira MS, da Cruz JN, Figueiredo PLB, Brasil DdSB, et al. (2019) Chemical profile of Lippia thymoides, evaluation of the acetylcholinesterase inhibitory activity of its essential oil, and molecular docking and molecular dynamics simulations. PLoS ONE 14(3): e0213393. https://doi.org/10.1371/journal.pone.0213393

• Angladon M-A, Fossépré M, Leherte L, Vercauteren DP (2019) Interaction of POPC, DPPC, and POPE with the μ opioid receptor: A coarse-grained molecular dynamics study. PLoS ONE 14(3): e0213646. https://doi.org/10.1371/journal.pone.0213646

• Schuster KD, Mohammadi M, Cahill KB, Matte SL, Maillet AD, Vashisth H, et al. (2019) Pharmacological and molecular dynamics analyses of differences in inhibitor binding to human and nematode PDE4: Implications for management of parasitic nematodes. PLoS ONE 14(3): e0214554. https://doi.org/10.1371/journal.pone.0214554

• Turner M, Mutter ST, Kennedy-Britten OD, Platts JA (2019) Molecular dynamics simulation of aluminium binding to amyloid-β and its effect on peptide structure. PLoS ONE 14(6): e0217992. https://doi.org/10.1371/journal.pone.0217992

• Miguel V, Villarreal MA, García DA (2019) Effects of gabergic phenols on the dynamic and structure of lipid bilayers: A molecular dynamic simulation approach. PLoS ONE 14(6): e0218042. https://doi.org/10.1371/journal.pone.0218042

• Woerner P, Nair AG, Taira K, Oates WS (2019) Sparsification of long range force networks for molecular dynamics simulations. PLoS ONE 14(4): e0213262. https://doi.org/10.1371/journal.pone.0213262

Reviewer #2: The authors present a molecular dynamics simulation study exploring the effect of electric field pulses on the superoxide dismutase protein. There are two main motivations for this study as presented by the authors: (i) to understand the structural stability of the protein in respect to an external electric field and (ii) to explore the possibility that the external electric field modifies the electrostatic guidance, i.e., the local electric potential that drives the superoxide to Cu2+ for the catalytic electron-transfer reaction. The goal (i) was already studied in a number of previous papers and the authors confirm the basic conclusion that sufficiently strong fields lead to partial protein denaturation. Question (ii) is potentially more novel and I believe here the authors’ analysis was insufficient.

The authors explore the distribution of the electric field, and its modification by an external pulse, in the vicinity of the active site. The first question to ask here: why should one care about the electric field? The ability of the superoxide to reach the active site should be driven by the potential of mean force, a free energy, and not by the electric field, which provides the force at a local point. I believe the authors are not calculating the property which is critical for the question they have posed. 

The referee asked a very interesting question that we do not properly addressed in the previous version of the manuscript. Our analysis was driven by previous works by D’Alessandro et al. and Amadei et al. [M. D'Alessandro, M. Aschi, M. Paci, A. Di Nola and A. Amadei Theoretical modeling of enzyme reactions chemistry: the electron transfer of the reduction mechanism in CuZn Superoxide Dismutase. J. Phys. Chem. B 108(41) 16255-16260 (2004); A. Amadei, M. D'Alessandro, M. Paci, A. Di Nola and M. Aschi On the Effect of a Point Mutation on the Reactivity of CuZn Superoxide Dismutase: A Theoretical Study. J. Phys. Chem. B 110(14),7538-7544 (2006)] where authors combined mixed QM/MM methods with basic statistical mechanical relations to study the chemical events and the atomic motions of the complex environment of the SOD1 reaction center. Their results clearly showed that the protein-solvent environment fluctuations are essential to understand the reaction mechanism which is based on the concerted rupture of the copper-histidine coordination bond and the copper-superoxide bond in the active site. Such environmental fluctuations have been conceived as a perturbation electric potential exerted by the environment on the quantum center, hence the sum of each elementary electric potential produced by i) water molecules solvating the protein; ii) counterions; iii) all protein atoms not belonging to the reactive center (treated via QM methods).

Amadei et al. concluded that such perturbing environmental electric field is essential for the modelling of the reaction process, pointing out dramatic effect of the protein and solvent interactions on free energy surface at the quantum center.

In the present work, we wanted to ask the question if an exogenous electric perturbation (in addition to the endogenous one) can significantly modify the electric environment at the quantum center, tackling the problem with a classical approach, i.e. established that the reaction free energy is affected by the local electric field at the active site.

I also have significant questions regarding how the electric field was calculated. The calculation is based on eq 1, which is the Coulomb law in vacuum. The resulting numbers are based on the magnitudes of partial charges and the distances to them. This is clearly not the entire picture. Electrostatic interactions are screened by water. Near the protein surface one cannot simply used the bulk dielectric constant of 78 and one has to specifically calculate the electric field by the water dipoles. However, the crudest estimate suggests that the numbers presented by the authors are overestimated by a factor of ~78. This is clearly not acceptable. In addition, I assume simulations were done in the standard Ewald protocol. Therefore, Ewald corrections have to be used in the calculation of the electric field as well. The calculations presented by the authors have no physical meaning unless these problems are addressed.

The referee is right; we just presented an oversimplified version of the perturbing field calculation. What we actually did is now presented in more details in the new version of the manuscript. Such perturbing field is an atomic electric field (on the order of GV/m), made by three terms: the first one representing the perturbation due to each aminoacidic residue of the protein, the second which considers the perturbation due to local electric field generated by water molecules, the last one due the presence of couterions. 

The new Eq. 1 now explicitly takes into account the contribution of all the constituents of the simulation box, as also explained in [Francesca Apollonio, Andrea Amadei, Micaela Liberti, Massimiliano Aschi, Monica Pellegrino, Maira D'Alessandro, Marco D'Abramo, Alfredo Di Nola, Guglielmo d'Inzeo Mixed Quantum-Classical Methods for Molecular Simulations of Biochemical Reactions with Microwave Fields: the Case Study of Myoglobin. IEEE T Microw. Theory 56(11), 2511-2519 (2008)] for a different protein, with all simulation data already taking into account Ewald corrections. Also note that MD systems correspond to needle-like ellipsoidal systems with the applied field along the major axis, hence no depolarizing field is present in the simulations.

We hope the issue raised by the referee is now clarified, the electric field perturbation at the active site was explicitly calculated at full atomistic scale.

It would be useful to have a consistency check for the electrostatic calculations. Can pKa be calculated to make sure the results are solid? None of the plots presented in the paper are testable by observations. The authors should make some minimum effort to connect to the observable reality.

The point raised by the referee drove us to calculate pKa values for some specific residues at the active site known to be fundamental for the reaction process. 

D’Alessandro et al. [M. D'Alessandro, M. Aschi, M. Paci, A. Di Nola and A. Amadei Theoretical modeling of enzyme reactions chemistry: the electron transfer of the reduction mechanism in CuZn Superoxide Dismutase. J. Phys. Chem. B 108(41) 16255-16260 (2004)] already investigated the endogenous electric perburbation effects on such residues, evidencing the relevance of each residue for the catalytic process. In general, residues producing a positive average electric field projected along the Cu-O2- bond favour the electron transfer process, while negative average electric fields are associated to inhibitory effects. 

Then, we analysed via PROPKA algorithm [Improved treatment of ligands and coupling effects in empirical calculation and rationalization of pKa values. Søndergaard CR, Olsson MHM, Rostkowski M, Jensen JH. J Chem Theory Comput. 2011;7(7):2284-95] the specific residues indicated in [A. Amadei, M. D'Alessandro, M. Paci, A. Di Nola and M. Aschi On the Effect of a Point Mutation on the Reactivity of CuZn Superoxide Dismutase: A Theoretical Study. J. Phys. Chem. B 110(14),7538-7544 (2006)] to affect the electronic transfer process.

We added in the new version of the manuscript a new Figure (Fig. 8), where we present pKa data of four fundamental residues: i) His 63 and His 120, directly involved in the reaction process; ii) Asp 83, which is known to favour the electron transfer; iii) His 71, which is known to inhibit the electron transfer. Data are presented together with their standard deviations, obtained collecting data of the simulation equilibrium states (150 ns before pulse application, last 50 ns of the bipolar pulse and 50 ns after the bipolar pulse removal).

Results indicate a significant effect on pKa values for all the considered residues (p-values have been calculated between reference and field-exposed populations), suggesting that a single high intensity (10^8 V/m) bipolar pulse is able to affect the local environment at SOD1 reactive site. In particular, negative variations of pKa values are associated to the build-up of a negative charge on residue side chain. Interestingly, such effects are consistent with the ones presented in Fig. 7, where the electric field shifts (absolute values) due to the applied Bipolar pulse was supposed to be a consequence of possible rotation of specific protein residues, and hence a charge distribution alteration.

All the above discussion and clarifications have been added in the main text.

---

## [Editor Report · Decision Letter 1]

14 Aug 2019

Nanosecond pulsed electric signals can affect electrostatic environment of proteins below the threshold of conformational effects: the case study of SOD1 with a molecular simulation study

PONE-D-19-16769R1

Dear Dr. Apollonio,

I do not like purely theoretical works, conclusion of which are not checked experimentally but the rules of PLoS One allow publishing such studies. For this reason:

We are pleased to inform you that your manuscript has been judged scientifically suitable for publication and will be formally accepted for publication once it complies with all outstanding technical requirements.

With kind regards,

Eugene A. Permyakov, Ph.D., Dr.Sci.

Academic Editor

PLOS ONE
---

## [Editor Report · Acceptance letter]

20 Aug 2019

PONE-D-19-16769R1 

Nanosecond pulsed electric signals can affect electrostatic environment of proteins below the threshold of conformational effects: the case study of SOD1 with a molecular simulation study 

Dear Dr. Apollonio:

I am pleased to inform you that your manuscript has been deemed suitable for publication in PLOS ONE. Congratulations! Your manuscript is now with our production department. 

With kind regards,

on behalf of

Prof. Eugene A. Permyakov 

Academic Editor

PLOS ONE